

# Co-administration of either curcumin or resveratrol with cisplatin treatment decreases hepatotoxicity in rats *via* anti-inflammatory and oxidative stress-apoptotic pathways

Osama I. Ramadan[1,2], Lashin S. Ali[3,4], Fatma M. Abd-Allah[2], Rafik E. A. Ereba[5], Humeda S. Humeda[6,7], Ahmed A. Damanhory[8,9], Ahmed E. Moustafa[10], Amr M. Younes[11], Moaaz M. Y. Awad[11,12] and Nassar A. A. Omar[12,13]

[1] Department of Dental Basic Sciences, Faculty of Dentistry, Applied Science Private University, Amman, Jordan
[2] Histology Department, Damietta Faculty of Medicine, Al-Azhar University, Cairo, Egypt
[3] Department of Basic Medical Science, Faculty of Dentistry, Al-Ahliyya Amman University, Amman, Jordan
[4] Department of Medical Physiology, Faculty of Medicine, Mansoura University, Mansoura, Egypt
[5] Department of Pharmacology, Faculty of Medicine, Al-Azhar University, Cairo, Egypt
[6] Department of Physiology, Faculty of Medicine, Alzaiem AlAzhari University, Khartoum North, Sudan
[7] Physiology Department, General Medicine Practice Program, Batterjee Medical College, Aseer, Saudi Arabia
[8] Department of Biochemistry, Faculty of Medicine, Al-Azhar University, Cairo, Egypt
[9] Department of Biochemistry, General Medicine Practice Program, Batterjee Medical College, Jeddah, Saudi Arabia
[10] Medical Biochemistry Department, Damietta Faculty of Medicine, Al-Azhar University, Cairo, Egypt
[11] Anatomy and Embryology Department, Damietta Faculty of Medicine, Al-Azhar University, Cairo, Egypt
[12] Anatomy Department, General Medicine Practice Program, Batterjee Medical Collage, Aseer, Saudi Arabia
[13] Department of Anatomy, Faculty of Medicine, Sohag University, Sohag, Egypt

Corresponding author
Osama I. Ramadan,
o_bayyoumi@asu.edu.jo

## ABSTRACT

**Background:** Cisplatin (CIS) is a broad-spectrum anticancer drug, with cytotoxic effects on either malignant or normal cells. We aimed to evaluate the hepatotoxicity in rats caused by CIS and its amelioration by the co-administration of either curcumin or resveratrol.

**Materials and Methods:** Forty adult male rats divided into four equal groups: (control group): rats were given a saline solution (0.9%) once intraperitoneally, daily for the next 28 days; (cisplatin group): rats were given a daily oral dose of saline solution (0.9%) for 28 days after receiving a single dose of cisplatin (3.3 mg/kg) intraperitoneally for three successive days; (CIS plus curcumin/resveratrol groups): rats received the same previous dose of cisplatin (3.3 mg/kg) daily for three successive days followed by oral administration of either curcumin/resveratrol solution at a dose of (20 mg/kg) or (10 mg/kg) consequently daily for 28 days. Different laboratory tests (ALT, AST, ALP, bilirubin, oxidative stress markers) and light microscopic investigations were done.

**Results:** Administration of CIS resulted in hepatotoxicity in the form of increased liver enzymes, oxidative stress markers; degenerative and apoptotic changes, the co-administration of CIS with either curcumin or resveratrol improved hepatotoxicity through improved microscopic structural changes, reduction in liver enzymes activity, decreased oxidative stress markers, improved degenerative, and apoptotic changes in liver tissues.

**Conclusion:** Co-administration of either curcumin or resveratrol with cisplatin treatment could ameliorate hepatotoxicity caused by cisplatin in rats *via* anti-inflammatory and oxidative stress-apoptotic pathways.

# INTRODUCTION

Cisplatin (CIS) is the first platinum-based medication licensed by the FDA to treat tumors, and it effectively works in curing a variety of solid tumors, such as breast, ovarian, bladder, and colon cancer (*Zhou et al., 2023*). It forms DNA adducts that prevent DNA replication and gene transcription in cancer cells, thereby exhibiting its anti-tumor properties (*Abo-Elmaaty et al., 2020*). Also, cisplatin has been used as the medication of choice for the treatment of cancer because of its mode of action involves DNA damaging in cancer cells by crosslinking purine bases in DNA, which induces apoptosis (*Alkhalaf, Mohamed & El-Toukhy, 2023*).

However, a major concern with antitumor chemotherapy is its lack of selectivity since cytostatic medications interact with both tumor cells and rapidly proliferating normal, healthy cells in the same way (*El-Gizawy et al., 2020*). Therefore, patients treated with chemotherapeutic drugs as cisplatin are susceptible to serious hazards as nephrotoxicity, cardiotoxicity, neurotoxicity, and hepatotoxicity which significantly impairs their clinical outcomes (*Tang et al., 2023*). Furthermore, hepatotoxicity was emerged as a significant adverse effect of CIS-based chemotherapy that limits its dosage (*Alkhalaf, Mohamed & El-Toukhy, 2023*).

The precise mechanism causing this damage is still unknown. However, several researches have suggested the cause of CIS-induced hepatotoxicity by the accumulation of CIS-metabolites in the liver leading to generation of reactive oxygen species (ROS) mediating oxidative stress-dependent mechanism (*Akcakavak, Kazak & Yilmaz Deveci, 2023*; *Louisa et al., 2023*). Moreover, oxidative stress-related cell death triggers an inflammatory response and is crucial to the pathophysiology of CIS-induced hepatotoxicity (*Aboraya et al., 2022*).

Herbal-based compounds are a subset of modern pharmacotherapy that cause fewer side effects in patients (*Chupradit et al., 2022*). Hence, we need to find natural or chemical bioactive compounds with antioxidant and anti-inflammatory effects to overcome CIS-induced hepatotoxicity. Several scientific researches have demonstrated that the use of antioxidants is highly significant to our bodies because of their tendency to scavenge and

stabilize free radicals, thereby preventing any cellular damage that may be caused by oxygen species and free radicals (*Al-Baqami & Hamza, 2021*).

There is a growing interest in plant-derived compounds as a result of the need to develop alternate sources of novel medications with advantageous health features. Naturally occurring products have been employed for the treatment and prevention of a number of chronic diseases, including cancer. Among the most varied classes of secondary metabolites found in plants, polyphenolic compounds are particularly well-known for their ability to serve as multifunctional agents in promoting health as potential anticancer and hepatoprotective agents (*Micale et al., 2021*).

Curcumin is the primary phenolic curcuminoid present in turmeric (curcuma longa) with a broad pharmacologic action which include antimicrobial, anticarcinogenic, anti-inflammatory, antioxidant, immunomodulatory, and antimutagenic properties (*Ali et al., 2020*). Prior researches have investigated that the application of curcumin as a hepatoprotective substance may enhance cisplatin anticancer action and lessen the adverse hepatic toxicity caused by the chemotherapeutic drugs (*Louisa et al., 2023*; *de Porras et al., 2023*; *Palipoch et al., 2014*).

Resveratrol (RSV) is a naturally occurring polyphenolic structure present in grapes and plums which exist as a monomer or as an oligomer with two to four monomer units (*Ibrahim et al., 2021*). Resveratrol is known to possess various health-promoting effects as anti-inflammatory, anti-oxidative, anti-neoplastic, and antimicrobial characters (*Inchingolo et al., 2022*). Several studies have investigated the efficacy of co-administration of RSV with cisplatin to augment its anticancer effect in the treatment of many carcinogenic diseases as ovarian, breast and uterine cancers and decrease the side effects on various body organs as kidney, liver, testis, and heart (*Wang et al., 2009*; *Abd-Elhafiz & Issa, 2021*; *Aly & Eid, 2020*).

Although, curcumin and resveratrol had low water solubility, poor absorption rates, limited bioavailability and enhanced oxidation upon exposure to heat and light, but, they have been the most investigated polyphenolic compounds in the last two decades, because of their many powerful medicinal effects, particularly their strong antioxidant and inflammatory effects (*Intagliata et al., 2019*).

Also, many studies revealed that the unique use of either resveratrol or curcumin had protective effects on alleviating cisplatin-induced damage in rat's liver or kidney (*El-Gizawy et al., 2020*; *Kara & Kilitci, 2022*). However, no research had studied the combined effects of either curcumin or RSV with cisplatin on the liver of rats in a single study.

Combination treatment and molecular hybridization are useful methods for enhancing the activity of polyphenolic compounds (curcumin and resveratrol) (*Hosseini-Zare et al., 2021*).

Moreover that, there were many deficiencies in the biochemical and histopathological parameters in assessment of the protective effects of those polyphenolic compounds used in the previous studies including, structural, anti-inflammatory, antioxidative, as well as the use of image analyzing program in assessment of the anti-fibrotic, anti-apoptotic effects.

Hence, we aimed to examine the potential protective effects of co-administration of either curcumin or RSV with cisplatin on the liver of rats by evaluating laboratory, histopathological, image analyzing data and immunohistochemical changes.

## MATERIALS AND METHODS

### Ethical approval

The ethical committee of the Damietta Faculty of Medicine, Al-Azhar University, Egypt, approved the experimental protocol (DFM-IRB 0001267-23-08-014) and all animals received the appropriate care followed the rules of national institutes of animal's health service policy on the use of laboratory animals.

### Chemicals and dosage

Cisplatin was purchased from Sigma-Aldrich as 1 mg per 1 ml concentration which was administered intraperitoneally by a single injection of cisplatin in a dose of 3.3 mg/kg every day for 3 days (*Alrashed & El-Kordy, 2019*). Curcumin was purchased from local market, (authenticated and identified by a specialist at the Botany department, Faculty of Agriculture, Al-Azhar University) in the form of dry rhizome which was mechanically ground and then extracted using boiling water for a full night. Three reports on the technique were made, and it involved pooling, concentrating under low pressure, and freeze drying (*Ahmed, El-Deib & Ahmed, 2010*). Following its suspension in 0.05% gum acacia solution, 20 mg/kg of curcumin was given orally (1 ml of the solution contained 2.5 mg of curcumin) (*Diab et al., 2014*).

Resveratrol was purchased from Sigma-Aldrich, produced freshly in 0.9% normal saline at a concentration of (10 mg/kg; daily) *via* oral gavage (*Ibrahim et al., 2021*).

### Animals

We used 10 animals/group to avoid any interference of results with any possible mortality rates and good statistical analysis of biochemical results.

Forty adult male albino rats (11 weeks age), weighing between 115 and 145 g each, they were obtained from Serum and Vaccine Institute at the Agricultural Research Center, Cairo, Egypt. They were housed in the animal house of the Damietta Faculty of Medicine, Al-Azhar University, Egypt, randomly assigned into groups, housed in a labelled hygienic, well-ventilated steel cages (5/cage) at room temperature and under carefully regulated light/dark cycles (12/12 h). They also had unlimited access to tap water and regular rodent chow a week prior to the experiment, to become acclimatized and all through the experiment.

### Experimental design

Rats were acclimatized in a room with a consistent temperature of 22 ± 1 °C, humidity (45–65%) and randomly separated into four groups (10 rats/group) and organized in a labeled cages (5/cage) as follows: Group I (Control): rats were given a saline solution (0.9%) once intraperitoneally, daily for the next 28 days; Group II (cisplatin only): rats were given a daily oral dose of saline solution (0.9%) for 28 days after receiving a single

dose of cisplatin (3.3 mg/kg) intraperitoneally for three successive days; Group III (cisplatin plus curcumin): rats were given a single dose of cisplatin (3.3 mg/kg) intraperitoneally daily for three successive days, then received a daily oral dose of curcumin solution (20 mg/kg) for 28 consecutive days; Group IV (cisplatin plus Resveratrol): rats were given a single dose of cisplatin (3.3 mg/kg) intraperitoneally daily for three successive days, then received a daily oral dose of resveratrol solution (10 mg/kg) for 28 consecutive days.

## Sampling

After 4 weeks from the first dose of cisplatin treatment, 4% isoflurane (SEDICO Pharmaceuticals, Cairo, Egypt) in 100% oxygen was used to anesthetize the rats, each rat's retro-orbital plexus was punctured to extract blood samples, which were then placed in sterile, dry centrifuge tubes and allowed to coagulate for 30 min at room temperature (RT) in a slanted position. The serum was then extracted by centrifuging the samples at $1,200 \times g$ for 20 min, and it was stored at $-20\ °C$ until it was needed for additional biochemical study.

Following the collection of blood, all groups of animals were sacrificed by cervical decapitation, and each rat's liver was removed before being cleaned with physiological saline, a specimen of liver tissue was processed for determination of oxidative stress, lipid peroxidation parameters and another specimen was fixed in $n$ 10% neutral buffered formalin for histological and immunohistochemical study. Outcomes were blindly assessed by the investigator who is ignorant of either treated or control rats. All expected or unexpected adverse events were recorded.

## Biochemical estimation

### The serum activity of liver enzymes

Alanine aminotransferase (ALT) (LOT: 32307166), aspartate aminotransferase (AST) (LOT: 10107023), alkaline phosphatase (ALP) (LOT: 32060283); and the serum levels of total and direct bilirubin (LOT: 202188) were measured by the kits provided by the Biodiagnostic Company (Cairo, Egypt). The indirect bilirubin levels were estimated using the difference between the total and direct bilirubin levels.

### Analysis of lipid peroxidation and antioxidant status in the liver tissue homogenate

A specimen of liver tissue was homogenized in Tris-HCl buffer (pH 7.4), after centrifuging the homogenate for 10 min at $4\ °C$ at 3,000 rpm, the supernatant was kept at $-20\ °C$ until determination of the oxidative stress parameters as glutathione (GSH), superoxide dismutase (SOD), catalase, and glutathione peroxidase (GPx), lipid peroxidation was also quantified in terms of malondialdehyde (MDA) production in liver tissue homogenates by the aid of the commercially available kits provided by Biodiagnostic Company (Cairo, Egypt) using spectrophotometry (Mispa Viva, Swiss) (*Ruiz-Larrea et al., 1994*).

### Assay of inflammatory markers

The serum levels of pro-inflammatory cytokines (tumor necrosis factor-α, interleukin-1β, and interleukin-6) and anti-inflammatory cytokines (interleukin-10) were measured by sandwich-based rat enzyme-linked immune sorbent assay (ELISA) kits provided by RayBiotech (Peachtree Corners, GA, USA).

### Histopathological assessment

Another liver specimen was fixed in 10% formaldehyde and further processed for histological analysis by a histologist blinded to the study groups using Raywild light microscope with built in camera (15 mega pixels) and image analyzing system after staining with hematoxylin and Eosin stain for structural changes. Masson trichrome stain for fibrotic changes & caspase three for apoptotic changes (*Suvarna, Layton & Bancroft, 2018*).

## Statistical analysis

The experimental findings were presented as means ± SD, or means ± standard deviation. All data were analyzed using the Statistical Package for the Social Sciences for Windows, version 20.0 (SPSS Inc., Chicago, IL, USA). Multivariate statistical analysis and ANOVA were used to compare the groups, and $P \leq 0.05$ was chosen as the significance probability.

## RESULTS

### Biochemical parameters

The cisplatin only group had a significant increase in the mean blood levels of ALT, AST, ALP, T. bilirubin, D. bilirubin, and I. bilirubin ($P < 0.05$) when compared to the control group, while the groups of co-administration of either curcumin or resveratrol with cisplatin treatment revealed a significant reduction in the previous parameters ($P < 0.05$) when compared to the cisplatin only group (Table 1).

The cisplatin only group had a significant reduction in the mean tissue levels of SOD, GPX and CAT ($P < 0.05$); and a significant elevation in the mean tissue level of MDA ($P < 0.05$) in comparison to the control group, while the groups of co-administration of either curcumin or resveratrol with cisplatin treatment revealed a significant increase in the mean tissue levels of SOD, GPX and CAT ($P < 0.05$); and a significant decrease in the mean tissue level of MDA ($P < 0.05$) in comparison to the cisplatin only group (Table 2).

The cisplatin only group had a significant elevation in the mean serum inflammatory markers levels (IL-1β, IL-6, and TNF-α) ($P < 0.05$) and a significant reduction in the level of IL-10 in comparison to the control group, while the groups of co-administration of either curcumin or resveratrol with cisplatin treatment revealed a significant reduction in the previous parameters ($P < 0.05$) and a significant increase in the level of IL-10 in comparison to the cisplatin only group (Table 3).

The morphometric analysis of the image analyzing study of the liver section stained with Masson trichrome for detection of collagen deposition (fibrosis) revealed a significant elevation ($P < 0.05$) in the percentage area of collagen deposition in the liver of rats exposed to cisplatin only compared to control group, while, there were significant decrease in the

**Table 1 Investigation of the levels of serum liver enzymes in the study groups.**

| Parameter/Group | Control | CIS | CIS+CUR | CIS+RES |
|---|---|---|---|---|
| ALT (U/L) | 28.9 ± 2.6 | 79.00 ± 12.35[*] | 46 ± 7.13[#] | 45.8 ± 5.39[#] |
| AST (U/L) | 26.7 ± 3.2 | 77.7 ± 11.68[*] | 42 ± 5.64[#] | 43.9 ± 5.13[#] |
| ALP (mg/dl) | 60.9 ± 8.56 | 115.2 ± 9.47[*] | 77.5 ± 9.07[#] | 74.1 ± 7.85[#] |
| T. bilirubin | 0.44 ± 0.03 | 0.71 ± 0.05[*] | 0.50 ± 0.02[#] | 0.49 ± 0.02[#] |
| D. bilirubin | 0.13 ± 0.01 | 0.24 ± 0.01[*] | 0.15 ± 0.02[#] | 0.16 ± 0.01[#] |
| I.bilirubin | 0.32 ± 0.03 | 0.47 ± 0.06[*] | 0.35 ± 0.03[#] | 0.34 ± 0.02[#] |

Notes:
ALT, Alanine transaminase; AST, Aspartate transaminase; ALP, alkaline phosphatase; Cis, Cisplatin; CUR, Curcumin; RES, Resveratrol; T. bilirubin, total bilirubin; D. bilirubin, direct bilirubin; I. bilirubin, indirect bilirubin.
[*] Significant differences between the CIS and control groups.
[#] Significant differences between CIS plus CUR or RES-treated groups and CIS group.

**Table 2 Investigation of the levels of oxidative/antioxidative stress markers in the study groups.**

| Parameter/Group | Control | CIS | CIS+CUR | CIS+RES |
|---|---|---|---|---|
| SOD (ng/ml) | 7.12 ± 0.96 | 3.19 ± 0.77[*] | 5.57 ± 0.70[#] | 5.26 ± 0.83[#] |
| GPX (ng/ml) | 6.63 ± 0.58 | 2.66 ± 0.36[*] | 4.07 ± 0.24[#] | 4.95 ± 0.53[#] |
| CAT (ng/ml) | 156.08 ± 6.58 | 79.64 ± 5.41[*] | 103.37 ± 8.15[#] | 116.12 ± 9.17[#] |
| MDA (ng/ml wet tissue) | 82.39 ± 8.72 | 281.43 ± 11.82[*] | 114.35 ± 4.83[#] | 107.21 ± 5.49[#] |

Notes:
SOD, Superoxide dismutase; GPx, glutathione peroxidase; CAT, catalase; MDA, malondialdehyde; $H_2O_2$, hydrogen peroxide.
[*] Significant differences between the CIS and control groups.
[#] Significant differences between CIS plus CUR or RES-treated groups and CIS group.

**Table 3 Investigation of the levels of serum inflammatory parameters in the study groups.**

| Parameter/Group | Control | CIS | CIS+CUR | CIS+RES |
|---|---|---|---|---|
| IL-1β (pg/ml) | 82.5 ± 8.41 | 232.83 ± 15.95[*] | 118.18 ± 17.68[#] | 109.69 ± 12.14[#] |
| IL-6 (pg/ml) | 1.29 ± 0.12 | 2.47 ± 0.2[*] | 1.84 ± 0.14[#] | 1.54 ± 0.13[#] |
| IL-10 (pg/ml) | 102.94 ± 6.29 | 48.2 ± 7.91[*] | 78.77 ± 8.89[#] | 73.33 ± 10.36[#] |
| TNF-α (ng/l) | 122.47 ± 11.18 | 235.33 ± 16[*] | 140.7 ± 9.87[#] | 148.57 ± 10.6[#] |

Notes:
IL-1β, Interleukin-1beta; IL-6, interleukin-6; IL-10, interleukin-10; TNFα, Tumor Necrosis Factor Alpha.
[*] Significant differences between the CIS and control groups.
[#] Significant differences between CIS plus CUR or RES-treated groups and CIS group.

collagen deposition in the groups of co-administration of either curcumin or resveratrol with cisplatin treatment when compared to the cisplatin only group (Table 4).

The morphometric analysis of the image analyzing study of the liver section stained with caspase-3 immune stain (apoptosis) revealed a significant elevation ($P < 0.05$) in the percentage area of caspase-3 immune stain expression in the liver of rats exposed to Cisplatin only compared to control group, while, there were significant decrease in the caspase-3 immune stain expression in the groups of co-administration of either curcumin or resveratrol with cisplatin treatment when compared to the cisplatin only group (Table 4).

**Table 4 Investigation of the Liver fibrosis and apoptosis parameters in the study groups.**

| Parameter/Group | | Control | CIS | CIS+CUR | CIS+RES |
|---|---|---|---|---|---|
| **Collagen** $(\mu m)^2$ | C.V region | $1.24 \pm 0.11$ | $3.48 \pm 0.16^*$ | $2.16 \pm 0.41^{\#}$ | $2.08 \pm 0.40^{\#}$ |
| | PT region | $1.32 \pm 0.14$ | $3.6 \pm 0.64^*$ | $2.69 \pm 0.21^{\#}$ | $2.33 \pm 0.26^{\#}$ |
| **Caspase-3** $(\mu m)^2$ | C.V region | $0.74 \pm 0.09$ | $3.01 \pm 0.44^*$ | $1.55 \pm 0.31^{\#}$ | $1.44 \pm 0.15^{\#}$ |
| | PT region | $0.8 \pm 0.08$ | $3.40 \pm 0.57^*$ | $1.73 \pm 0.35^{\#}$ | $1.49 \pm 0.16^{\#}$ |

Notes:
* Significant differences between the CIS and control groups.
\# Significant differences between CIS plus CUR or RES-treated groups and CIS group.

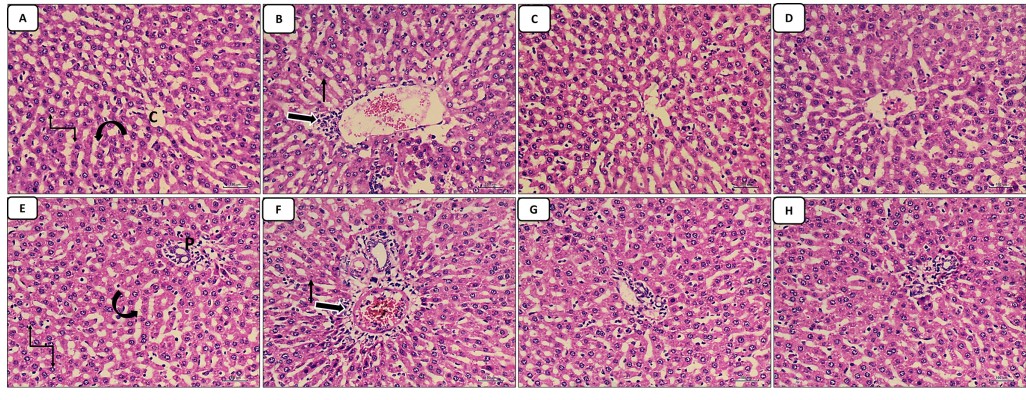

**Figure 1 (A–H) HX. & E.**

## Histopathological assessments

The liver sections stained with HX.&E: The control group appear with normal central vein, normal portal tract, hepatocytes arranged in cords and normal blood sinusoids in-between (Figs. 1A and 1E); the cisplatin group showed dilated congested central vein and portal tract in addition to infiltration of the central vein and periportal region with inflammatory cells, increased pyknotic cells (Figs. 1B and 1F). The groups of co-administration of either curcumin or resveratrol with cisplatin treatment revealed amelioration of the structural lesions caused by cisplatin-only in the liver tissue with more or less restoration of the diameters of central and portal veins (Figs. 1C–1H).

The liver sections stained with masson trichrome: the control group revealed minimal collagen deposition in both regions of the central vein and portal tract (Figs. 2A and 2E). The cisplatin-treated group revealed marked collagen deposition (marked fibrosis) around the central vein and portal vein in both regions of the central vein and portal tract, in addition to the marked dilatation and congestion in both the central vein and portal tract (Figs. 2B and 2F). The groups of co-administration of either curcumin or resveratrol with cisplatin treatment revealed amelioration of the collagen deposition caused by cisplatin-only in the liver tissue with more or less restoration of the diameter of the central vein and portal tract which appeared less congested than the cisplatin treated group (Figs. 2C–2H).

The liver sections stained with caspase-3 immune stain: the control group revealed minimal expression of the immune stain in both the central vein and portal tract regions,

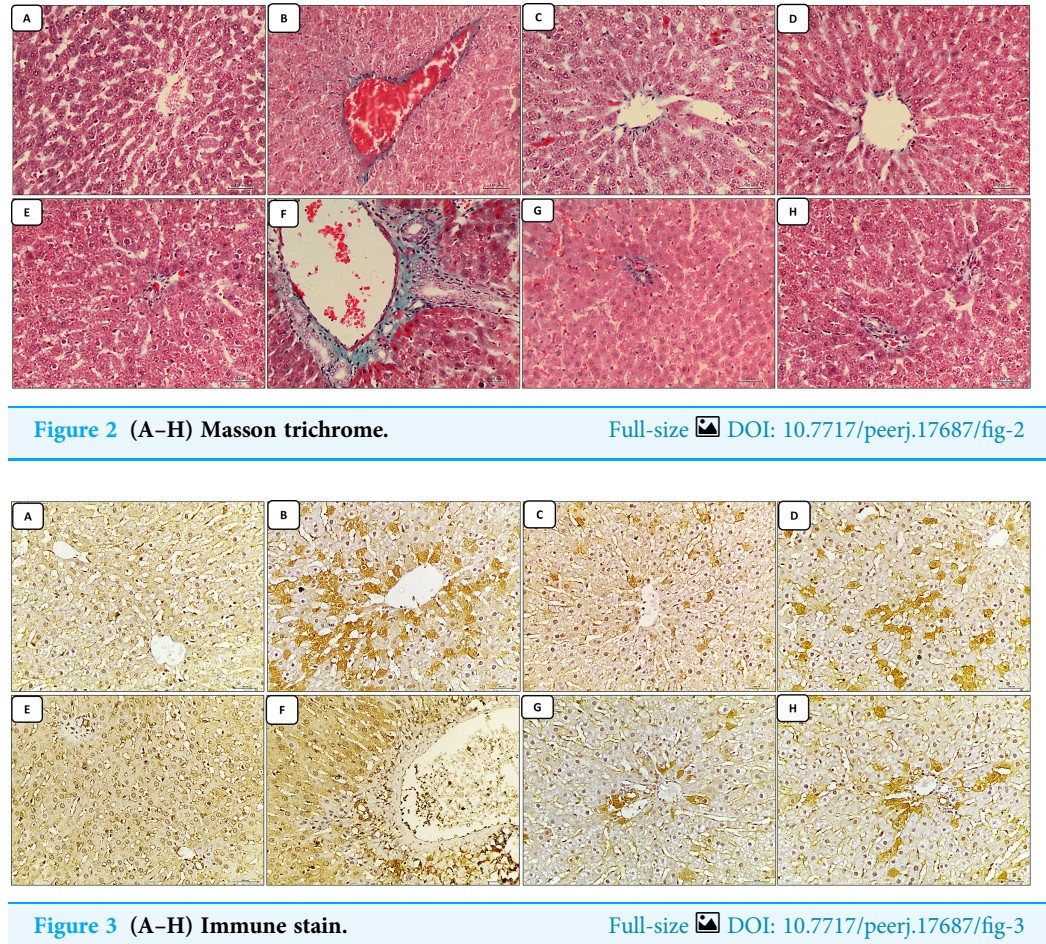

**Figure 2 (A–H) Masson trichrome.**

**Figure 3 (A–H) Immune stain.**

which appeared normal in structure (Figs. 3A and 3E). The cisplatin-treated group revealed marked expression of the caspase-3 immune stain (marked apoptosis) in both regions of the central vein and portal tract, in addition to excessive dilatation of central vein and portal vein (Figs. 3B and 3F). While, the groups of co-administration of either curcumin or resveratrol with cisplatin treatment revealed amelioration of expression of the immune stain induced by cisplatin-only in the liver tissue, also both of central vein and portal vein were restored to normal diameter compared to the cisplatin treated group (Figs. 3C–3H).

## DISCUSSION

Cisplatin (CIS) is one of the most potent cytotoxic anticancer drugs with hepatotoxic effects suggested to be caused by increased ROS production and cellular damage (*Aboraya et al., 2022*). Moreover, it results in induction of oxidative stress which can lead to inflammation and the synthesis of cytokines including TNF-α and IL-6. It has been shown that elevated ROS and pro-inflammatory cytokines cause hepatocyte apoptosis (*Ingawale, Mandlik & Naik, 2014*).

In this study, we assessed the effect of co-administration of either curcumin or resveratrol with cisplatin treatment to decrease the CIS-induced hepatotoxicity in rats *via* anti-inflammatory and oxidative stress-apoptotic pathways.

Administration of cisplatin to rats in a dose of 3.3 mg/kg b.wt. resulted in structural histopathological changes in the liver in the form of infiltration of the central vein and periportal region with inflammatory cells, dilated congested central vein and portal tract, increased pyknotic cells. This was similar to the findings of a previous studies documented that the liver sections of a rat exposed to CIS in different doses (1, 3.3, 5, 7.5,15 mg/kg b. wt.) displayed signs of liver damage, including sinusoidal dilatation, vascular congestion, inflammatory cell infiltration of the liver's stroma and portal triad, and focal sites of degeneration (*Aboraya et al., 2022*; *Ogbe, Agbese & Abu, 2020*; *Bademci et al., 2021*; *Qu et al., 2019*; *El-Sayyad et al., 2009*; *Pace et al., 2003*).

In this study, the alteration in the structural changes in the liver due to cisplatin exposure was corelated with the alteration in the liver functions as proved by the notable elevation in the mean blood levels of liver enzymes (ALT, AST, ALP) in the cisplatin exposed group in comparison to the control group. This was in agreements to the assay of liver functions results of other studies investigating the hepatotoxic effects of cisplatin exposure in rats (*Alkhalaf, Mohamed & El-Toukhy, 2023*; *Akcakavak, Kazak & Yilmaz Deveci, 2023*; *Ogbe, Agbese & Abu, 2020*).

The ability of CIS to cause an increase in ALT, AST, and ALP serum activity is thought to occur as a byproduct of CIS-induced liver injury and the subsequent hepatocyte leakage of these enzymes and may indicate liver degeneration and fibrosis (*Gressner, Weiskirchen & Gressner, 2007*).

Moreover, we assayed the bilirubin levels in our study as it is a well-known marker of tissue damage from toxic chemicals, the serum levels of T. bilirubin, D. bilirubin, I. bilirubin were elevated in the cisplatin exposed group in comparison to the control group. In accordance to our results a recent study (*Aboraya et al., 2022*) assumed that rats treated with CIS demonstrated hyperbilirubinemia as a result of higher levels of total and indirect bilirubin, they suggested the elevated indirect bilirubin levels caused by hemolytic anemia in the hematological picture. Also, it may have come from either the reduced rate of bilirubin conjugation in the liver or the decreased hepatic absorption of bilirubin (*VanWagner & Green, 2015*).

In contrary to our results a previous study revealed normal albumin level following CIS injection, despite a notable drop in the levels of total protein and globulin (*Neamatallah et al., 2018*). This could be due to the difference in rat ages, species, timing and mode of treatment.

This was in agreements to the assay of liver functions results of other studies investigating the hepatotoxic effects of cisplatin exposure in rats.

The deterioration in the liver structure and function due to exposure to cisplatin in our study could be caused by the induction of oxidative stress caused by the imbalance between oxidant-antioxidant levels influenced by increased generation of ROS (reactive oxygen species) in CIS-treated rats and reduce the scavenging power toward ROS. This was evidenced by the increase in hepatic MDA level and the decrease in enzymatic

antioxidants, such as hepatic SOD and CAT. Furthermore, the hepatic tissue of rats given CIS is more vulnerable to oxidative stress due to the reduction of GPX levels. Similar to our findings several researches revealed increased oxidative parameters (MDA) in the liver tissue of rats exposed to CIS, along with a decrease in the enzymatic antioxidant activity including liver tissue levels of CAT, SOD and GPX (*Bentli et al., 2013*; *Omar et al., 2016*).

In this study, oxidative stress in the liver induced by CIS-exposure has been predicted to cause inflammation as noticed by downregulation of IL-10 and significant upregulation of IL-1β, IL-6, TNF-α in the hepatic tissue, as well as apoptosis, which is confirmed by overexpression of hepatic caspase-3 immune stain. This was coinciding with results of a previous researches assayed the previous markers in rat's liver exposed to CIS (*Neamatallah et al., 2018*; *Tahoun, Elgedawy & El-Bahrawy, 2021*).

The protein TNF-α is linked to apoptosis and has a role in inflammatory responses and the IL-10 can also be released by apoptotic cells. Thus, our findings indicate that CIS-induced hepatotoxicity is linked to inflammatory and apoptotic pathways as the first stage in the onset of apoptosis brought on by a variety of triggers is caspase activation.

Moreover, oxidative stress generation in the cisplatin-treated group is responsible for the marked deposition of collagen fibers in the liver tissue indicating fibrosis that was seen in our study, which was similar to a previous study revealed a noticeable accumulation of collagen fibers (*He et al., 2006*).

The results of this research displayed that co-administration of either curcumin or resveratrol with cisplatin treatment decreases liver toxicity in rats as noticed by improvement in the liver structure (amelioration of pathological changes) and functions (ameliorations of liver enzymes and bilirubin levels).

Similar to our results multiple researchers found that co-administration of curcumin with cisplatin treatment was found to decreases hepatotoxicity induced by CIS in rats (*El-Gizawy et al., 2020*; *Palipoch et al., 2014*; *Ahmed, El-Deib & Ahmed, 2010*; *Diab et al., 2014*). The beneficial use of curcumin in this study to reduce CIS-mediated hepatotoxicity was suggested to be evidenced by its anti-inflammatory effect *via* its ability to eradicate the free radicals, reduce pro-inflammatory cytokine levels (TNF-α, IL-6, and IL-1β); antiapoptotic effect through reducing the immune-expression of caspase-3; and antifibrotic effects as stated by previous studies (*El-Gizawy et al., 2020*; *Louisa et al., 2023*; *He et al., 2006*; *Gao et al., 2022*).

Along similarity, multiple studies found that co-administration of resveratrol with cisplatin treatment was found to decreases hepatotoxicity induced by CIS in rats (*Ibrahim et al., 2021*; *Wang et al., 2009*; *Abd-Elhafiz & Issa, 2021*). The hepatoprotective effect of resveratrol in this study was suggested to be through its ability to reduce the pro-inflammatory cytokine levels (TNF-α, IL-6, and IL-1β); antiapoptotic effect through decrease in the expression of caspase-3; and antifibrotic effects as revealed by different studies (*Al-Baqami & Hamza, 2021*; *Abd-Elhafiz & Issa, 2021*; *Liu et al., 2018*).

From the above findings, the present study was the first study to compare the co-administration of either resveratrol or curcumin with cisplatin through different techniques in a single study to prove the potential protective effects of both compounds

and found that both compounds have antifibrotic, anti-inflammatory, antiapoptotic, antioxidative effects ameliorating the hepatoxic effects of cisplatin.

## CONCLUSIONS

Co-administration of either curcumin or resveratrol with cisplatin treatment could ameliorate hepatotoxicity caused by cisplatin in rats *via* anti-inflammatory and oxidative stress-apoptotic pathways in spite of the low absorption rates of either resveratrol or curcumin.

### Limitations of the study

The combination therapy used in this study gave limited predictions about whether or not real cancer patients will take it as a co-medication. Also, the low absorption rates of either resveratrol or curcumin appears to limit their *in vivo* biological effects which represent a major barrier in the development of therapeutic applications for the compounds. Hence, a newer delivery system of those polyphenolic compounds including nanoparticles is suggested to be investigated in the future studies.

### Funding
The authors received no funding for this work.

### Competing Interests
The authors declare that they have no competing interests.

### Author Contributions
- Osama I. Ramadan conceived and designed the experiments, performed the experiments, analyzed the data, prepared figures and/or tables, authored or reviewed drafts of the article, and approved the final draft.
- Lashin S. Ali conceived and designed the experiments, performed the experiments, analyzed the data, prepared figures and/or tables, authored or reviewed drafts of the article, and approved the final draft.
- Fatma M. Abd-Allah conceived and designed the experiments, performed the experiments, analyzed the data, prepared figures and/or tables, authored or reviewed drafts of the article, and approved the final draft.
- Rafik E. A. Ereba conceived and designed the experiments, analyzed the data, prepared figures and/or tables, authored or reviewed drafts of the article, and approved the final draft.
- Humeda S. Humeda conceived and designed the experiments, analyzed the data, prepared figures and/or tables, authored or reviewed drafts of the article, and approved the final draft.
- Ahmed A. Damanhory conceived and designed the experiments, analyzed the data, prepared figures and/or tables, authored or reviewed drafts of the article, and approved the final draft.

- Ahmed E. Moustafa conceived and designed the experiments, performed the experiments, analyzed the data, prepared figures and/or tables, authored or reviewed drafts of the article, and approved the final draft.
- Amr M. Younes conceived and designed the experiments, performed the experiments, analyzed the data, prepared figures and/or tables, authored or reviewed drafts of the article, and approved the final draft.
- Moaaz M. Y. Awad conceived and designed the experiments, performed the experiments, analyzed the data, prepared figures and/or tables, authored or reviewed drafts of the article, and approved the final draft.
- Nassar A. A. Omar conceived and designed the experiments, analyzed the data, prepared figures and/or tables, authored or reviewed drafts of the article, and approved the final draft.

## Animal Ethics

The following information was supplied relating to ethical approvals (*i.e.*, approving body and any reference numbers):

The ethical committee of the Damietta Faculty of Medicine, Al-Azhar University, Egypt approved the study (DFM-IRB 0001267-23-08-014).

## Data Availability

Raw data are available in the Supplemental Files.

## Supplemental Information

Supplemental information for this article can be found online at http://dx.doi.org/10.7717/peerj.17687#supplemental-information.

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
