# Peer review of "Co-administration of either curcumin or resveratrol with cisplatin treatment decreases hepatotoxicity in rats via anti-inflammatory and oxidative stress-apoptotic pathways"

_PeerJ, doi:10.7717/peerj.17687_

## Round 0.1 · original submission · Minor Revisions

Dear authors, i apologise for the delay in my decision.

Your work requires some revisions before publication. Please, address the reviewers' comments. Also, improve your methods section and proofread carefully.

Indicate catalogue number/reference in relevant materials such as cisplatin, resveratrol, transaminases kits, etc...!

Mention clearly the rationale for your experimental design, namely in terms of dosages. and make sure that all your figures have the required dpi/resolution for publication. all the methods used for the results should be mentioned in the appropriate section. and make sure that figure legends are ADEQUATE and complete, indicating the clear highlight of that figure/table.

Many thanks.

Reviewer 1 ·

Basic reporting

This paper explores the effect of co-administration of curcumin or resveratrol with cisplatin on rats to ameliorate related effects. Although interesting, the number of animals per group (10 rats/group) is high according to 3Rs. Histological pictures are of good quality.

Experimental design

The experimental design is suitable. However missing data were noted:
Masson trichrome staining was not mentioned in Methods

Validity of the findings

Results are interesting and well discussed

Additional comments

Other comments:
L. 29 - anticancer
L. 59 – Please remove “in normal tissues,”
L. 67 – Herbal-based compounds instead of Herbal biological substances
L. 69 – hepatotoxicity
L. 70 – What is the meaning of: “excessive consumption of antioxidants is highly significant?
L. 73 - Curcuma longa
L.77 - cisplatin anticancer
L. 92 – Please correct: health's public health
L. 100 - was given instead of was taken
Please include humidity values for animal house
L. 130 - all groups of animals
L. 135 – Please remove :in results
L. 156 – Please use full name of stainings H&E
Masson trichrome staining was not mentioned in Methods
L. 273 – Please cite more references
L.275 - by its

·

Basic reporting

no comment

Experimental design

no comment

Validity of the findings

no comment

Additional comments

no comment

Reviewer 3 ·

Basic reporting

The English usage in this passage could be enhanced with more elaboration. Upon reviewing the manuscript, I found similarities to previously published works, such as those referenced in the citations 10.4274/haseki.galenos.2022.8098 and 10.1007/s00210-020-01888-0. These studies also focused on single compounds like resveratrol or curcumin.

However, a key question arises: why combine these compounds when it's widely recognized that both curcumin and resveratrol have poor absorption rates? This warrants further clarification.

Experimental design

I don't see any novelty in this experimental design

Validity of the findings

I don't see any novelty in this findings works

---

## Round 0.2 · Minor Revisions

Dear authors, one of the reviewers presented relevant concerns that have not been properly addressed! Not in the rebuttal and certainly absolutely not included in the manuscript itself. Thus, please refer to the similarities / differences between previous studies and justify the need for this study on the manuscript itself. Also, think through and improve (and include in the document) why combining the compounds specially since both are widely studied worldwide (even CDC, gov USA has special research groups in herbal products like this) and have been proven difficult to standardize results. Transparency and clarity in your message may be critical to convey a message. I will consider publication of your work upon the addressing of these points.

---

## Round 0.3 · Minor Revisions

Dear authors,

While the study you presented holds promise, several key points still require clarification and improvements before it can be considered for publication - pardon if i have missed something in the manuscript.

Firstly, I'd like to address the rationale behind combining compounds with known poor absorption rates, such as curcumin and resveratrol. It's crucial to clearly justify this approach, especially considering the established challenges regarding their absorption. Without a compelling rationale, readers may question the scientific merit of the study. Therefore, I kindly request you to elaborate on this aspect in your manuscript. In fact, highlight it.

Furthermore, the manuscript still lacks proper legends for the figures and comprehensive legends for the tables. Legends should not only describe the content of the figures and tables but also highlight key results and observations. In particular, I noted that some histopathology images (Figure 2 and 3) lack marked key observations. Please ensure that all figures and tables are accompanied by detailed legends that adequately summarize their findings.

Moreover, the methodological approach, particularly regarding the acquisition of curcumin, raises concerns about standardization. I recognise that this is a major obstacle to herbal drugs. But do not run from this, rather you can mention limitations. Consider discussing the study's implications in the larger context of herbal drug standardization and study design.

Furthermore, the manuscript should address the limitations of the study and provide insights into future perspectives. Given the significant focus on mitigating hepatotoxicity via anti-inflammatory and oxidative stress-apoptotic pathways, it's imperative to acknowledge the limitations inherent in the study design and methodology. Additionally, providing insights into future research directions will enhance the manuscript's overall impact and relevance.

Lastly, I must highlight that in this case, it's really essential to clearly articulate the value proposition of studying compounds with poor absorption rates. What unique insights or advancements does this research offer to the field? Clarifying the significance and potential applications of your findings will strengthen the manuscript's contribution to the scientific community. The sentence used does not do this job well enough.

In conclusion, I encourage you to address these concerns and incorporate the suggested improvements into your manuscript. While the novelty and utility may be questioned, the way you convey your message may surpass the shortfalls.

---

## Round 0.4 · Major Revisions

Dear authors, unfortunately, i do not think you addressed the shortcomings mentioned in my last decision. I will give you one last opportunity to address things properly.

Not even the figure/table legends have been completed and properly describing the data, and that is not acceptable! Also remember that any particular kits, antibodies, pharma products, machinery, etc must be properly and completely described which includes model/catalog number.

For example, the justification you gave about not following the 3Rs for animal research, i did not find it in any way mentioned in the manuscript itself. and the same for the responses you gave about the studies highlighted by one of the reviewers: 10.4274/haseki.galenos.2022.8098 & 2- 10.1007/s00210-020-01888-0. You should have made a better effort and mention even more studies than those highlighted, in the manuscript itself. Furthermore, conclusions and limitations can and should be improved. Another important detail is in line 157-158... i think you mean "enhancing polyphenolic compounds properties, or decreasing its biological use limitations? THUS, language is important. as it was highlighted before. I copy paste here my last decision for your consideration! :

Firstly, I'd like to address the rationale behind combining compounds with known poor absorption rates, such as curcumin and resveratrol. It's crucial to clearly justify this approach, especially considering the established challenges regarding their absorption. Without a compelling rationale, readers may question the scientific merit of the study. Therefore, I kindly request you to elaborate on this aspect in your manuscript. In fact, highlight it.

Furthermore, the manuscript still lacks proper legends for the figures and comprehensive legends for the tables. Legends should not only describe the content of the figures and tables but also highlight key results and observations. In particular, I noted that some histopathology images (Figure 2 and 3) lack marked key observations. Please ensure that all figures and tables are accompanied by detailed legends that adequately summarize their findings.

Moreover, the methodological approach, particularly regarding the acquisition of curcumin, raises concerns about standardization. I recognise that this is a major obstacle to herbal drugs. But do not run from this, rather you can mention limitations. Consider discussing the study's implications in the larger context of herbal drug standardization and study design.

Furthermore, the manuscript should address the limitations of the study and provide insights into future perspectives. Given the significant focus on mitigating hepatotoxicity via anti-inflammatory and oxidative stress-apoptotic pathways, it's imperative to acknowledge the limitations inherent in the study design and methodology. Additionally, providing insights into future research directions will enhance the manuscript's overall impact and relevance.

Lastly, I must highlight that in this case, it's really essential to clearly articulate the value proposition of studying compounds with poor absorption rates. What unique insights or advancements does this research offer to the field? Clarifying the significance and potential applications of your findings will strengthen the manuscript's contribution to the scientific community. The sentence used does not do this job well enough.

External reviews were received for this submission. These reviews were used by the Editor when they made their decision, and can be downloaded below.

---

## Round 0.5 · accepted · Accept

Dear authors,

I believe we are reaching a point where further discussion will not be productive and issues can be addressed during the proofreading stage. Your discussion is particularly weak, specifically in addressing the problem with the polyphenols. However, your data can support other researchers seeking phytochemicals to mitigate the secondary effects of chemotherapy through personalized delivery strategies or other methods to improve delivery, bioavailability, and efficacy.
Therefore, I have decided to proceed with this decision. It is concerning, however, that you misunderstood reference or catalog numbers of transaminases (and other materials) for lot numbers, which are essential indeed but for in-house quality controls. How can I reproduce your experiments based on your descriptions? Imagine if the kits provided by Biodiagnostic Company do not adhere to any ISO or quality parameters and mistakenly measure AST, SOD and catalase. What would that do to this entire work?

I strongly urge you to use the English editing services from PeerJ if you do not have a native speaker or qualified colleague who can help with the language. The manuscript is still very weak in this aspect, and science also relies on clear communication.

Please, ensure you present a high-quality final document for production!